# Effect of Selenium Biofortification and Beneficial Microorganism Inoculation on Yield, Quality and Antioxidant Properties of Shallot Bulbs

**DOI:** 10.3390/plants8040102

**Published:** 2019-04-17

**Authors:** Nadezhda Golubkina, Svetlana Zamana, Timofei Seredin, Pavel Poluboyarinov, Sergei Sokolov, Helene Baranova, Leonid Krivenkov, Laura Pietrantonio, Gianluca Caruso

**Affiliations:** 1Federal Scientific Center of Vegetable Production, Selectsionnaya str. 14, VNIISSOK, Odintsovo district, 143072 Moscow, Russia; timofey-seredin@rambler.ru (T.S.); elena-shevcovabaranova@mail.ru (H.B.); krivenkov76@mail.ru (L.K.); 2State University of Land Management, Kazakova str. 15, 10506 Moscow, Russia; svetlana.zamana@gmail.com; 3Penza State University of Architecture and Construction, Titova str. 28, 440028 Penza, Russia; poluboyarinovpavel@yandex.ru; 4Scientific Technical Center ‘Sustainable Development of Agroecosystems’, 143072 Moscow, Russia;sergey.alex.sokolov@gmail.com; 5Department of Agricultural Sciences, University of Naples Federico II, 80055 Portici, Naples, Italy; laura.pietrantonio77@gmail.com (L.P.); gcaruso@unina.it (G.C.)

**Keywords:** *Allium cepa* L. *Aggregatum* group, biofortification, selenocystine, sodium selenate, arbuscular mycorrhizal fungi

## Abstract

Plant biofortification with selenium in interaction with the application of an arbuscular mycorrhizal fungi (AMF)-based formulate, with the goal of enhancing Se bioavailability, is beneficial for the development of the environmentally friendly production of functional food with a high content of this microelement. Research was carried out in order to assess the effects of an AMF-based formulate and a non-inoculated control in factorial combination with two selenium treatments with an organic (selenocystine) or inorganic form (sodium selenate) and a non-treated control on the yield, quality, antioxidant properties, and elemental composition of shallot (*Allium cepa* L. *Aggregatum* group). Selenocystine showed the best effect on the growth and yield of mycorrhized plants, whereas sodium selenate was the most effective on the non-inoculated plants. The soluble solids, total sugars, monosaccharides, titratable acidity, and proteins attained higher values upon AMF inoculation. Sodium selenate resulted in higher soluble solids, total sugars and monosaccharide content, and titratable acidity than the non-treated control, and it also resulted in higher monosaccharides when compared to selenocystine; the latter showed higher protein content than the control. Calcium, Na, S, and Cl bulb concentrations were higher in the plants inoculated with the beneficial microorganisms. Calcium and sodium concentrations were higher in the bulbs of plants treated with both the selenium forms than in the control. Selenocystine-treated plants showed enhanced accumulation of sulfur and chlorine compared to the untreated plants. The AMF inoculation increased the bulb selenium content by 530%, and the Se biofortification with selenocystine and sodium selenate increased this value by 36% and 21%, respectively, compared to control plants. The AMF-based formulate led to increases in ascorbic acid and antioxidant activity when compared to the non-inoculated control. The bulb ascorbic acid was increased by fortification with both selenium forms when compared to the non-treated control. The results of our investigation showed that both AMF and selenium application represent environmentally friendly strategies to enhance the overall yield and quality performances of shallot bulbs, as well as their selenium content.

## 1. Introduction

The shallot (*Allium cepa* L. *Aggregatum* group) is one of the most widespread *Allium* species across the world, and its bulbs are characterized by high levels of biological activity, including antioxidant [1], hypoglycemic [2], hypocholesterolemic [3], antibacterial and antifungal, and anticoagulation [4] activity. In this respect, this vegetable is an excellent candidate for the health-centered strategy of producing functional foods with high levels of Se and antioxidants. Indeed, selenium (Se) is an essential element, showing powerful antioxidant properties and thus protecting the human organism against cardiovascular diseases and cancer [5]. Unfortunately, the uneven distribution of Se over the Earth’s surface causes wide Se-deficient areas, leading to increased risk of these mentioned health issues. The similarity in chemical properties between Se and sulfur triggers the processes of sulfur replacement by Se in biological systems, resulting in the formation of Se-containing amino acids, proteins, carbohydrates, and other biologically active compounds [6]. Biofortification with Se is considered an effective approach for tackling the human Se deficiency status [7], and plant accumulation of this microelement may be enhanced by arbuscular mycorrhizal fungi (AMF). The latter encourage plant growth and development, as they positively affect nutrient uptake efficiency and biomass, soil pathogen suppression, and tolerance of drought, salinity, and heavy metals by producing phytohormones and changing the root morphology [8,9]. The inoculation of beneficial microorganisms represents an environmentally friendly tool which is gaining importance in modern agriculture in the improvement of vegetable yield and quality while decreasing the use of mineral fertilizers, herbicides, and insecticides [8,10].

Controversial results have been published about the effects of AMF on selenium accumulation in plants. Indeed, Larsen et al. [11] reported an increase of Se content in garlic by supplying AMF solely or in combination with sodium selenate, whereas the application of the AMF *Glomus claroideum* resulted in a 23% Se content increase in wheat [12]. Conversely, mycorrhizal inoculation reduced the Se content in lettuce [13], as well as in maize, alfalfa, and soybean [14].

In this respect, the *Allium* genus may draw special interest as it, along with *Brassica*, belongs to the secondary Se accumulators able to synthesize predominantly methylated forms of Se-containing amino acids and peptides (selenomethyl selenocysteine and γ-glutamyl selenomethyl selenocysteine) with a pronounced anticarcinogenic effect [15,16]. Moreover, among agricultural crops, the *Allium* species are particularly sensitive to the presence of arbuscular mycorrhizal fungi in the soil, which increase the root absorption surface through their hyphae, thus encouraging soil nutrient accumulation and water exploitation. AMF inoculation in *Allium* crops can also enhance plant antioxidant properties [10].

The presence of Se bioavailable forms in soil depends to a large extent on the presence of microorganisms in the rhizosphere which take part in redox reactions between Se^+6^, Se^+4^, Se^−2^, and Se^o^ [17], resulting both in protection against the toxic effect of Se and in improvement of the accumulation of this element in plants. 

Due to the scant data available about the effects of AMF and selenium application on the Se biofortification of *Allium cepa* L. *Aggregatum* group and to the controversial results relevant to the effects of AMF on plant selenium biofortification, further investigations regarding the prospects of this approach are needed. The present research aimed to assess the interactions between inoculation with AMF and the supply of organic or inorganic Se forms to shallot plants in terms of the root mycorrhizal colonization, bulb yield, quality, antioxidant properties, and elemental composition.

## 2. Material and Methods

### 2.1. General Methods

Research was conducted on shallot (*Allium cepa* L. *Aggregatum* group) cultivar Alba at the experimental fields of the Federal Scientific Center of Vegetable Production, Moscow region, Russia (55° 39′ 23″ N, 37° 12′ 43″ E) in 2017 and 2018. The trial was carried out on a clay-loam soil with pH 6.8, 2.1% organic matter, 108 mg·kg^−1^ N, 450 mg·kg^−1^ P_2_O_5_, 357 mg·kg^−1^ K_2_O, and an exchangeable base sum as high as 95.2%. The average air temperature values, recorded at plant level, were 13.5 °C in May, 17.1 °C in June, and 20.1 °C in July.

The experimental protocol was based on the factorial combination ofan arbuscular mycorrhizal fungi (AMF)-based formulate (Rhizotech Plus at 2 g·m^−2^ soil) and a non-inoculated control, two selenium treatments (sodium selenate at 63 mg Se ·m^−2^, by 50 mg·L^−1^ 0.26 mM solution;selenocystine, at the same dose as sodium selenate), and a non-treated control.

The AMF-based formulate Rhizotech MB (Msbiotech S.p.A., Larino, Campobasso, Italy) is a plant-growth-stimulating preparation which predominantly contains the endomycorrhizal fungus *Rhizophagus intraradices*, along with low concentrations of *Trichoderma harzianum* and *Bacillus subtilis*.

A split plot design with three replicates was used for the treatment distribution in the field, with the AMF inoculation assigned to the main plot and each elementary plot covering a 15 m^2^ (5 × 3 m) surface area. 

Shallot bulbs were planted on 5 and 4 May in 2017 and 2018, respectively, and bulbs were spaced 7 cm apart along single rows 35 cm apart (41 plants per m^2^).

The AMF-based formulate Rhizotech MB powder was appliedin each hole made in the soil prior to bulb planting. The treatments with selenium were repeated twice on 8 and 22 June, both in 2017 and 2018, by spraying the shallot plants with the aforementioned water solution. 

The mycorrhizal index, intended as the percentage of root mycorrhizal colonization, was assessed twice, one month after transplant and at the crop cycle end, according to Giovannetti and Mosse’s method [18].

Harvest was performed in the last week of July in both research years when the bulbs had reached their maximum growth, and in each plot the following determinations were made: number and weight of bulbs; mean bulb weight from50-unit samples. In addition, a 25 marketable bulb sample was randomly collected in each plot and transferred to the laboratory in order to perform analytical determinations, 

### 2.2. Sample Preparation

After harvesting, bulbs were separated from the plants, cleaned, removed from the outer shells, and cut with a plastic knife to form thin slices. Fresh homogenized material was used for nitrate and ascorbic acid determination. An aliquot of fresh slices wasdried at 70 °C to constant weight and used for the determination of polyphenols, flavonoids, and antioxidant activity. The Se content in bulbs was determined in homogenized samples dried at 20 °C to constant weight. 

### 2.3. Dry Residue and Soluble Solids

The dry residue was assessed in an oven at 70 °C until constant weight. The soluble solids (°Brix) were determined at 20 °C using a digital refractometer by Bellingham and Stanley, model RFM 81, on the supernatant obtained by centrifuging the raw homogenate.

### 2.4. Sugars

The monosaccharides were determined using the ferricyanide colorimetric method based on the reaction of monosaccharides with potassium ferricyanide [19]. The total sugars were analogically determined after acidic hydrolysis of water extracts with 20% hydrochloric acid. Fructose was used as an external standard.

### 2.5. ElementalComposition

The Al, As, B, Ca, Cd, Co, Cr, Cu, Fe, Hg, I, K, Li, Mg, Mn, Na, Ni, P, Pb, Si, Sn, Sr, V, and Zn contents of the shallot bulbs were assessed using an ICP-MS on a quadruple mass spectrometer Nexion 300D (Perkin Elmer Inc., Shelton, CT 06484, USA) equipped with the 7-port FAST valve and ESI SC DX4 autosampler (Elemental Scientific Inc., Omaha, NE 68122, USA) in the Biotic Medicine Center (Moscow, Russia). Rhodium 103 Rh was used as an internal standard to eliminate instability during measurements. Quantitation was performed using an external standard (Merck IV, multi-element standard solution), potassium iodide for iodine calibration, and Perkin-Elmer standard solutions for P, Si, and V. All the standard curves were obtained at five different concentrations.

For quality checking purposes, internal controls and reference materials were tested together with the samples on a daily basis. Microwave digestion of samples was achieved with sub-boiled HNO_3_ diluted 1:150 with distilled deionized water (Fluka No. 02, 650 Sigma-Aldrich, Co., Saint Louis, MO, USA) in the Berghof SW-4 DAP-40 microwave system (BerghofProducts + Instruments Gmb H, 72, 800 Eningen, Germany). 

The instrument conditions and acquisition parameters were as follows: plasma power and argon flow, 1500 and 18 L min^−1^, respectively; aux argon flow, 1.6 L min^−1^; nebulizer argon flow, 0.98 L min^−1^; sample introduction system, ESI ST PFA concentric nebulizer and ESI PFA cyclonic spray chamber (Elemental Scientific Inc., Omaha, NE 68122, USA); sampler and slimmer cone material, platinum; injector, ESI Quartz 2.0 mm I.D.; sample flow, 637 L min^−1^; internal standard flow, 84 L min^−1^; dwell time and acquisition mode, 10–100 ms and peak hopping for all analytes; sweeps per reading, 1; reading per replicate, 10; replicate number, 3; DRC mode, 0.55 L min^−1^ ammonia (294993-Aldrich Sigma-Aldrich, Co., St. Louis, MO 63103, USA) for Ca, K, Na, Fe, Cr, V, optimized individually for RPa and RPq; STD mode, for the rest of analytes at RPa = 0 and RPq = 0.25.

### 2.6. Nitrates

Nitrates were assessed using an ion selective electrode by ionomer Expert-001 (Econix, Russia). Five grams of fresh shallot bulb homogenate were homogenized with 50 ml of distilled water. A quantity of 45 ml of the resulting extract wasmixed with 5 ml of 0.5 M potassium sulfate background solution (necessary for regulating the ionic strength) and analyzed through the ionomer for nitrate determination.

### 2.7. Selenium

Se was analyzed using the fluorimetric method previously described for tissues and biological fluids [20]. Dried homogenized samples were digested via heating with a mixture of nitric and chloral acids, subsequent reduction of selenate (Se^+6^) to selenite (Se^+4^) with a solution of 6 N HCl, and formation of a complex between Se^+4^ and 2,3-diaminonaphtalene. Calculation of the Se concentration was achieved by recording the piazoselenol fluorescence value in hexane at 519 nm λ emission and 376 nm λ excitation. Each determination was performed in triplicate. The precision of the results was verified using a reference standard of lyophilized cabbage in each determination with a Se concentration of 150 μg·Kg^−1^.

### 2.8. Antioxidants

#### 2.8.1. Ascorbic Acid

The ascorbic acid contentwas determined by visual titration of plant extracts in 6% trichloracetic acid with Tillmans reagent [21]. Three grams of fresh shallot bulb homogenates were homogenized in a porcelain mortar with 5 ml of 6% trichloracetic acid and quantitatively transferred to a measuring cylinder. The volume was brought to 60 ml using trichloracetic acid, and the mixture was filtered through filter paper 15 min later. The concentration of ascorbic acid was determined from the amount of Tillmans reagent that went into titration of the sample. 

#### 2.8.2. Polyphenols

Polyphenols were determined in water extract using the Folin–Ciocalteu colorimetric method as previously described [22]. One gram of dry shallot bulb powder was extracted with 20 ml of 70% ethanol at 80 °C over 1 h. The mixture was cooled and quantitatively transferred to a volumetric flask, and the volume was adjusted to 25 ml. The mixture was filtered through filter paper, and 1 ml of the resulting solution was transferred to a 25 ml volumetric flask to which 2.5 ml of saturated Na_2_CO_3_ solution and 0.25 ml of diluted (1:1) Folin–Ciocalteu reagent were added. The volume was brought to 25 ml with distilled water. One hour later the solutions were analyzed through aspectrophotometer (Unico 2804 UV, USA), and the concentration of polyphenols was calculated according to the absorption of the reaction mixture at 730 nm. As an external standard, 0.02% gallic acid was used.

#### 2.8.3. Flavonoids

The total flavonoids content was determined by a spectrophotometric method based on flavonoid–aluminum chloride (AlCl_3_) complexation [23]. The standard of quercetin dehydrate (98% HPLC) was purchased from Sigma Co. (St. Louis, MO, USA). A quantity of 10mL of methanol wasadded to 1 g of dried and homogenized samples and left at room temperature for 2 h. The resulting mixture was filtered through pleated filter. Then, 0.2 mL of the extract was diluted with 1.8 mL of methanol, and 0.1 mL of 2% AlCl_3_, 0.5 mL of 1 M sodium acetate solution, and 1 mL of distilled water were added. The mixture was incubated for 30 min at room temperature and the absorption at 415 nm was measured. The total flavonoid content was determined from astandard curve built using five different concentrations of quercetin–AlCl_3_ complex. Quercetin was purchased from Fluka (Switzerland).

#### 2.8.4. Antioxidant Activity (AOA)

The antioxidant activity of the shallot bulbs was assessed using a redox titration method [24] via titration of 0.01 N KMnO_4_ solution with ethanolic extracts from shallot bulbs. The reduction of KMnO_4_ to colorless Mn^+2^ in this process reflects the quantity of antioxidants dissolvable in 70% ethanol. The values were expressed in mg gallic acid equivalents (GAE)·100 g^−1^d.w. The use of KMnO_4_ acidic solution has been successfully used for the determination of the *Ocimum basilicum* antioxidant potential [25] and the antioxidant capacity of serum [26].

### 2.9. Statistical Analysis

Data were processed by two-way analysis of variance, and mean separations were performed through Duncan’s multiple range test, with reference to a 0.05 probability level, using SPSS software version 21. Data expressed as percentage were subjected to angular transformation before processing. 

## 3. Results and Discussion

All the variables analyzed in this research were not significantly affected by the year of research; therefore, only the mean data of the two years areshown. 

### 3.1. Mycorrhizal Index

The mycorrhizal index did not significantly change between the two determinations performed one month after the transplant and at the end of crop cycle; therefore, the mean values of the two determinations are reported in Table 1. This index was significantly affected by the AMFinoculation (Table 1), showing 65.7% mycorrhizal colonization in inoculated plant roots compared to 26.4% in the non-inoculated control. Differently to this, the selenium treatment did not have a significant effect on the mycorrhizal index, though selenocystine showed a better tendency to promote plant–microorganism symbiosis. This may partly be explained by the higher ability of selenocystine when compared with selenate to participate in redox reactions affected by AMF. AMF are known to release organic compounds, increasing bacterial density and accelerating microbial metabolic activity and nutrient cycling in the rhizosphere [27].

### 3.2. Plant Growth and Yield

As reported in Table 1, the leaf area (expressed as LAI) expanded both in the plants inoculated with mychorrizae and in those treated with selenocystine. Neither the mycorrhizal nor the selenium application significantly affected the number of bulbs per plant. Instead, the two experimental factors showed significant interactions on plant dry matter and on yield and mean bulb weight (Figure 1, Figure 2 and Figure 3). Indeed, the microorganism inoculation encouraged shallot dry matter accumulation, which was 51.6% higher compared to thenon-inoculated control, as an average of the selenium treatments. Consequently, the bulb yield was increased by 1.5 times on average, compared to the non-inoculated control, due to the higher mean bulb weight. With regard to selenium treatment of plants inoculated with AMF, the inorganic form (sodium selenate) showed a better effect than the non-treated control, whereas the organic compound (selenocystine) did not differ from the inorganic form or control either on growth or yield variables. Differently from this, the non-inoculated plants were best affected by the selenocystine treatment, with no significant difference between sodium selenate and the control. 

The shallot root system isbeneficially colonized by the two AMF genera *Glomus* and *Scutellospora*, with *Glomus* predominating [28]; this explains the significant effect of the mycorrhizal formulates in enhancing shallot plant growth and development in our research. The consequent increase in bulb yield upon AMF inoculation is in agreement with the previous reports [10]. In *Allium cepa*, AMF application also increased bulb yield by 2–18 timesdepending on the amount of phosphorus in the soil [29].

As for the Se supply, the decrease inshallot bulb number in the plants treated with selenocystine suggests that the organic form of this microelement promoted the formation of fewer but heavier bulbs, whereas the bulbs were not affected by the Se inorganic compound in terms of either weight or number (Table 1). Moreover, the yield reaction of shallot plants to the Se form applied was significantly affected by mycorrhizae which valorized the Se inorganic compound, but selenocystine showed effective stimulation of bulb production in the absence of beneficial microorganism inoculation. In previous research [30], sodium selenate application to soil during potato plant growth also decreased the number of tubers butincreased their weight. The decrease in the shallot bulb number may suggest amodified allocation pattern of assimilates as a consequence of selenocystine treatment; a similar effect was associated with inorganic Se in potato [30].

### 3.3. Quality Indicators

As reported in Table 2, the dry matter content in shallot bulbs was not affected by either the microorganism inoculation or selenium treatment. The soluble solids, total sugars, monosaccharides, titratable acidity, and proteins attained higher values upon AMF inoculation when compared to the non-inoculated control. As for the selenium treatments, sodium selenate resulted in higher soluble solids, total sugars and monosaccharide content, and titratable acidity than the non-treated control; Se^+6^ applicationalso led toa higher monosaccharide concentration compared to Se^−2^, but these did not differ with respect to titratable acidity. A higher content of total proteinsresulted in the treatment with the Se organic form than in the control (Table 2).

The effect of AMF on quality indicators of the *Allium* species, i.e., on the increase of sugar accumulation and antioxidant activity, has also been reported in previous research [10]. The beneficial microorganisms modulate biosynthesis and sugar concentration, in particular through the action of phytohormones [31,32]. 

Based on the results of our research, the positive effect of Se on yield (Table 1) may be due to the enhanced carbohydrate metabolism and, to some extent, to the antioxidant effects of this microelement. 

The ripeness process in *Allium cepa* and *Allium sativum* is accompanied by the increase of sucrose content and the decrease of glucose and fructose [33]; indeed, this sucrose biosynthesis may be correlated with the metabolism of these two monosaccharides and with the decomposition of other carbohydrates. The microorganismeffect may be connected with changes in the activity of different enzymes participating in the process of ripening, elicited bygrowth-stimulating substances [34]. 

The results are in agreement with the positive effect of AMF on the nutritional quality of edible plant parts recorded by other authors: rhizosphere microorganisms were found to enhance protein and carbohydrate accumulation by plants [8,35], and an increase inproteins, carotenoids, and flavonoids was recorded in lettuce upon AMF inoculation [36]. 

### 3.4. Elemental Composition

As can be observed in Table 3, higher concentrations of Ca, Na, S, and Cl were observed in the bulbs of shallot plants inoculated with AMF. With regard to selenium treatments, higher concentrations of calcium and sodium were observed in the bulbs of plants treated with either selenium form than in the control; Mg and Pwere not affected by Se treatments; and treatment with selenocystine enhanced the accumulation of sulfur and chlorine in plants compared to thatin the untreated plants.

Statistically significant interactions between microorganism inoculation and selenium treatment were recorded on the content of potassium, magnesium, phosphorus, and nitrates in shallot bulbs (Table 4). Indeed, potassium was better affected by (SeCys)_2_ compared to the control in the non-inoculated plants, whereas both Se forms resulted in K increase under AMF application; moreover, sodium selenate led to enhanced bulb potassium content in mycorrhized plants in comparison with the same Se treatment in non-inoculated plants. NO_3_ accumulation was encouraged by plant biofortification with both Se forms; in addition, the plants inoculated with beneficial microorganisms showed higher bulb nitrate content than the non-inoculated ones only under Se treatment. 

The average 27.7% increase in nitrate concentration under the AMF formulate application compared tonon-inoculated plants is presumably connected with the significant increase in shallot bulb weight. Indeed, nitrogen is known to be a key component for improving the yield and quality of *Allium* species [37], and AMF reportedly enhances nitrogen assimilation [38]. AMF inoculation in field-grown tomato plants resulted in a nitrate content increase, consistent with the beneficial microorganism effect found in previous research [31]. Kučová [39] reported that AMF decrease nitrate leaching from soil. 

The increased nitrate concentration in shallot bulbs may also be connected with the modulation of nitrate absorption as a result of AMF treatment, which is in accordance with the data published by Copetta et al. [40]. Studies of growing rates suggest that AMF can actively transfer nitrogen to the host plants, which stimulate this transport by supplying carbonto microorganisms [41]. Taking into account that the consumption of high-nitrate-content food may cause the formation of carcinogenic N-nitrosamines [42], the contents of this ion in shallot bulbs recorded in the present work show no risks for human organisms, referring to the threshold reported by European Committee guidelines no. 1881/2006 [43]. 

Although Se supply usually activates nitrate reductase, causing a decrease in plant nitrate levels, the latest studies on spinach Se biofortification suggest that the process is strictly connected with phytohormone participation, with contrasting results in female and male plants [22]. These findings may suggest the participation of shallot phytohormones in conditions of AMF and Se application. 

With regard to the microelements, the trace levels of Hg, Sn, As, Cd, Pb, and Ni in the samples were not taken into account. Moreover, the contents of Co, Cr, I, Li, Sr, and V in the shallot bulbs were not affected by the experimental treatments, and they are therefore not reported.

The inoculation of shallot plant roots with AMF increased the bulb selenium content by 530% and Se biofortification with (SeCys)_2_ and sodium selenate increased this value by 36% and 21%, respectively, compared to control plants (Table 5).

The interaction between microorganism inoculation and Se treatment was statistically significant on the content of B, Cu, Fe, Mn, Si, and Zn in shallot bulbs (Table 6). B and Fe were not affected by selenium treatment in non-inoculated plants, but their content dropped upon Se application in mycorrhized plants; moreover, B was higher in mycorrhized plants compared to non-inoculated ones when selenium was not applied, whereas Fe content was always higher in the inoculated plants. As for Cu and Zn, selenium application was not effective on plants inoculated with beneficial microorganisms, whereas sodium selenate or (SeCys)_2_ led to a higher concentration of Cu or Zn, respectively, compared to the non-Se-treated control; however, the beneficial microorganism inoculation resulted in higher Cu content not only in non-Se-treated plants but also under the Se treatments for Zn content. Se treatments enhanced Mn accumulation in non-inoculated plants, whereas the Se organic form caused a reduction of this element in mycorrhized plants; the latter showed higher Mn content than non-inoculated plants only in the non-Se-treated control. The Si content was reduced by sodium selenate application in non-inoculated plants and by both Se forms in mycorrhized plants; the latter always showed higher Si content than non-inoculated plants.

The application of AMF formulates to shallot plants resulted in the following average increases in microelements: 64.7% B, 47.4% Cu, 141% Fe, 77.9% Mn, 44.7% Zn, and 78.4% Si. These elements are essential for plant growth and development as they are involved in physiological processes. Indeed, Fe and Mn participate in redox reactions promoting photosynthesis; Cu, Mn, Zn, and Fe are important components of many enzymes; and Si encourages plant growth and development, increases B accumulation, and decreases the toxicity of heavy metals. The high levels of Si in AMF-inoculated plants detected in our research are consistent with the effect of AMF on increased Si content which is known to reduce Al accumulation [44]. Taking into account the high phytotoxicity of Al [45], the reduction of this element’s concentration as a result of AMF inoculation and Se treatment (Figure 4) contributed to stimulating shallot growth in our trial, consistent with the findings of Zhang et al. [46] relevant to the positive effect of mycorrhizae in preventing Al accumulation in plants. As far as human nutrition is concerned, high levels of B, Cu, Fe, Mn, Zn, and Si provide additional benefits to the use of shallot bulbs treated with AMF due to the essentiality of these elements to human beings.

The effect of AMF on Se accumulation is of remarkable interest. Previous investigations [11,12] showed the possibility of increasing plant Se concentration by applying AMF or a combination of AMF and sodium selenate to soil. The more intense effect of Se on shallot plants in our research compared to that recorded in wheat [12] may be connected with the higher reactivity of *Allium* species to AMF [10]. This phenomenon may be of practical importance, as it entails the chance of using environmentally friendly technologies for the increase of soil Se bioavailability due to the expansion of root surface promoted by AMF. In our research, the intensity of the mentioned achievement was higher withthe organic Se form (selenocystine) than with sodium selenate. The plant enrichment with Se provided much higher concentrations of this microelement, corresponding to 48 and 36 biofortification levels (for selenocystine and sodium selenate, respectively) compared to control plants. The lower concentration of Se in the bulbs of shallot plants treated with sodium selenate than in those treated with selenocystine is consistent with the reports relevant to the higher intensity of Se accumulation under the application of an organic Se form compared to inorganic forms [47]. Nevertheless, it should be pointed out that, to date, selenocystine has never been used for plant biofortification, despite its easier and cheaper chemical synthesis compared to selenomethionine [48]. Literature data show that decreasing Se accumulation upon Se supply occurs according to the following order: Se^+4^ < Se^+6^ < SeMet [47]. The results of the present study suggest the significance of AMF inoculation in enhancing the Se level in *Allium* species. In a previousinvestigation [49], the Se content in the roots and sprouts of Indian mustard was increased by rhizosphere microorganisms solely orin combination with sodium selenate treatment. 

In our research, the amount of Se ingested from 50 g of shallot bulbs treated with AMF formulates and selenocystine meets the optimal Se requirement, whereas 50 g of bulbs from plants sprayed with sodium selenate provides 63% of the daily Se requirement.

The correlation coefficients between the mineral elements analyzed are reported in Table 7. These data point out the relationships between macro- and microelements in shallot bulbs, highlighting the changes in the contents of themain elements caused by AMF inoculation and Se supply. The most significant positive correlations are B with Mn and P; Cu with Mn, Mg, and P; Fe with Si; Mn with Mg and P; Zn with Ca; Ca with NO_3_; K with Na and NO_3_; and Mg with P and Al.

### 3.5. Antioxidants

As reported in Table 8, both the polyphenol and flavonoid concentrations in shallot plants were not significantly affected by application of the AMF-based formulate, but the formulate led to an increase in ascorbic acid content and antioxidant activity compared to the non-inoculated control. As for the selenium treatments (Table 8), the polyphenol and flavonoid concentrations and antioxidant activity were not significantly affected, whereas the ascorbic acid content increased in the bulbs obtained from the plants biofortified with either selenium compound compared to the non-treated control. 

At present, the mechanisms concerning AMF, Se, and antioxidant interactions in plants are not clear. Several papers suggest that root AMF colonization can induce the accumulation of secondary metabolites, polyphenols, flavonoids, and vitamins in lettuce [36], potato [50], *A. cepa*, and leek [51]. Other reports have shown the absence of a significant effect on the antioxidant properties in lettuce and tomato [52,53]. This phenomenon may be connected with the biological dilution effect, caused by the significant increase in plant biomass and described for several plant species inoculated with AMF [52], or relatively low stress loading, which seems to reduce the positive effect of AMF [53]. 

Ascorbic acid affects mitosis and cell growth in plants, and it is an important cofactor for several enzymes involved in the synthesis of a number of secondary metabolites including phytohormones and in protection against environmental stresses [54]. In our trial, the highest content of ascorbic acid in shallot as a result of the application of both AMF and sodium selenate corresponded to the highest bulb yield and highest monosaccharides content (Table 1, Table 2 and Table 3).

Shallot is characterized by high antioxidant activity due to the presence of sulfur compounds (diallyl disulfide, diallyl trisulfide, allyl and allicin), flavonoids (glucosides of quercetin), polyphenols, ascorbic acid, and saponins [55,56]. The mean values of flavonoids and polyphenols in the present work were within the range recorded in Polish, Indonesian, and Vietnamese varieties [56,57].

Notably, the application of an AMF formulate provided a significant increase in plant antioxidant activity in our research, in accordance with previous reports related to *A. cepa* [51].

One of the possible factors contributing to these properties is the close relationship between AMF, Se, and phytohormones. 

It is known that hormones participate in AMF–host plant symbiosis and in antioxidant biosynthesis. In particular, gibberellins and auxins are intensive regulators promoting the synchronization of AMF spore formation and host plant growth [58,59]. 

Furthermore, plant biofortification with inorganic Se is also strictly connected with the plant hormonal status; this is reflected in the increase of female forms of hemp as a result of sodium selenite application [60] and higher selenate accumulation by male forms of spinach than by female ones, the former showing a high content of gibberellins [22]. Berhow [61] found that gibberellins are able to decrease the flavonoid content in grapefruit, and sodium selenate can reduce the quercetin content in shallot bulbs (Table 3). Gibberellins can also increase the quantity of monosaccharides and ascorbic acid content in onion [34], whereas high levels of these compounds were recorded in shallot bulbs in our study upon AMF inoculation and sodium selenate treatment (Table 2). However, additional investigations are necessary to reveal the role of phytohormones in relation to AMF and Se in plants.

## 4. Conclusions

The results from research carried out in northern Europe suggest interesting application prospects of the use of formulates based onarbuscular mycorrhizal fungi (AMF) in interaction with selenium biofortification by organic or inorganic compounds in shallot crops. Indeed, AMF inoculation resulted in higher production, whereas the selenium treatment effect depended on the compound form: sodium selenate positively affected bulb yield only when applied to mycorrhized plants, whereas selenocystine was only effective on non-inoculated plants. The values of bulb quality indicators, macro- and microelements, ascorbic acid, and antioxidant activity increased upon AMF inoculation; the two selenium forms positively affected most of the quality attributes and macroelements, as well as selenium and ascorbic acid, but their effect on the remaining microelements was unclear. The use of AMF-based formulates has proved to be a cost-effective and ecologically friendly method for enhancing the yield, nutritional properties, and selenium content of shallot bulbs.

## Figures and Tables

**Figure 1 plants-08-00102-f001:**
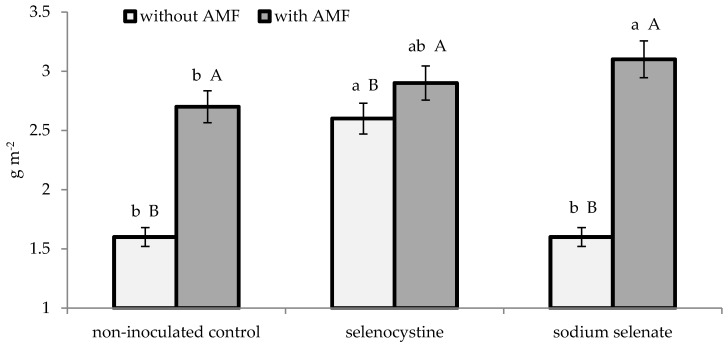
Interaction between microorganism inoculation and selenium treatmenton shallot dry matter. The lowercase letters refer to comparison between the selenium treatments within the AMF formulate and control, while capital letters refer to comparison between the AMF formulate and control within each selenium treatment; this is according to Duncan’s test at *p* ≤ 0.05 with three replicates per treatment.

**Figure 2 plants-08-00102-f002:**
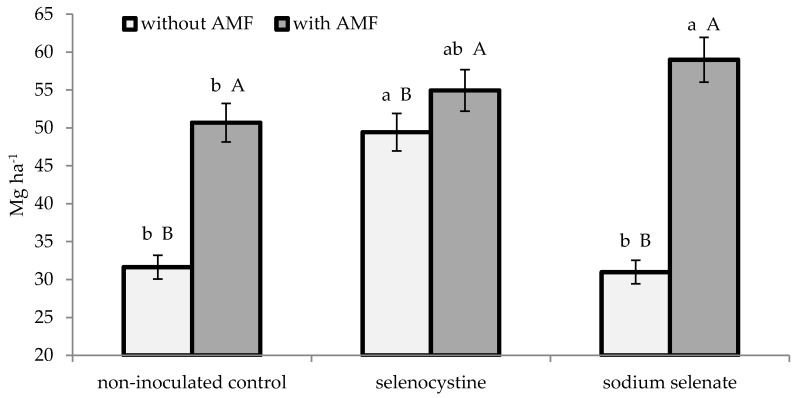
Interaction between microorganism inoculation and selenium treatment on shallot bulb yield. The lowercase letters refer to comparison between the selenium treatments within the AMF formulate and control, while capital lettersrefer to comparison between the AMF formulate and control within each selenium treatment; this is according to Duncan’s test at *p* ≤ 0.05 with three replicates per treatment.

**Figure 3 plants-08-00102-f003:**
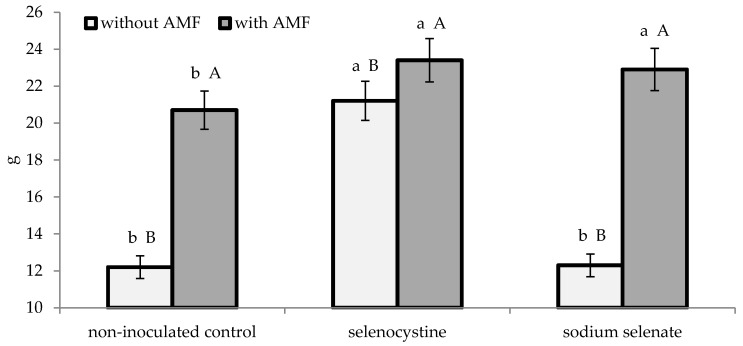
Interaction between microorganism inoculation and selenium treatment on shallot mean bulb weight. The lowercase letters refer to comparison between the selenium treatments within the AMF formulate and control, while capital letters refer to comparison between the AMF formulate and control within each selenium treatment; this is according to Duncan’s test at *p* ≤ 0.05 with three replicates per treatment.

**Figure 4 plants-08-00102-f004:**
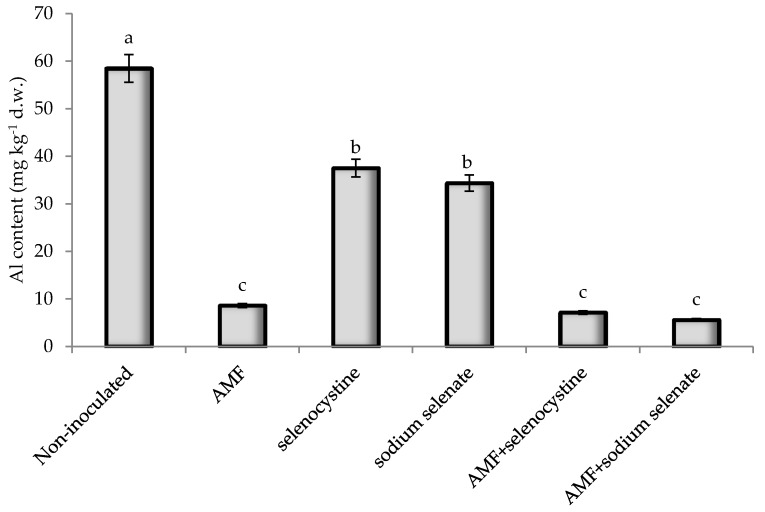
Aluminum content in shallot bulbs as affected by microorganism inoculation and selenium treatment. Values followed by different letters are statistically different according to Duncan’s test at *p* ≤ 0.05 with three replicates per treatment.

**Table 1 plants-08-00102-t001:** Effect of microorganism inoculation and selenium treatment on shallot bulb yield.

Experimental Factor	Mycorrhizal Index	Growth Indices				Marketable Bulbs				
	%	LAI (m^2^·m^−2^)		Dry Matter (g·m^−2^)		Yield (Mg·ha^−1^)		Mean Weight (g)		Number per Plant
Microorganism inoculation										
AMF formulate	65.7 ± 2.6	1.25 ± 0.11		2.91 ± 0.23		54.9 ± 2.4		22.3 ± 1.4		6.0 ± 0.3
Non-inoculated control	26.4 ± 1.1	0.82 ± 0.06		1.92 ± 0.15		37.3 ± 1.7		15.2 ± 0.9		6.1 ± 0.3
	*	*		*		*		*		n.s.
Selenium treatment										
Selenocystine	47.5 ± 1.8	1.18 ± 0.11	a	2.77 ± 0.23	a	52.2 ± 2.1	a	22.3 ± 1.5	a	5.8 ± 0.3
Sodium selenate	46.0 ± 1.5	1.00 ± 0.07	b	2.34 ± 0.16	b	45.0 ± 1.7	b	17.6 ± 1.1	b	6.3 ± 0.4
Non-treated control	44.7 ± 1.6	0.92 ± 0.07	b	2.14 ± 0.13	b	41.2 ± 2.3	b	16.5 ± 1.2	b	6.2 ± 0.3
	n.s.									n.s.

LAI, leaf area index; n.s. not significant; * significant at *p* ≤ 0.05. Within each column, values followed by different letters are statistically different according to Duncan’s test at *p* ≤ 0.05. Abbreviations: LAI, leaf area idex; AMF, arbuscular mycorrhizal fungi.

**Table 2 plants-08-00102-t002:** Shallot bulb quality indicators as affected by microorganism inoculation and selenium treatment.

Experimental Factor	DM (%)	SS (°Brix)		TS (g·100 g^−1^d.w.)		MS(%)		TA mg Citric Acid·100 g^−1^d.w.		TP (g·100 g^−1^d.w.)	
Microorganism inoculation											
AMF formulate	18.6 ± 0.06	15.3 ± 0.07		70.2 ± 0.39		10.2 ± 0.1		2.10 ± 0.08		10.2 ± 0.30	
Non-inoculated control	18.2 ± 0.06	14.4 ± 0.08		64.5 ± 0.67		8.4 ± 0.1		1.77 ± 0.03		9.3 ± 0.23	
	n.s.	*		*		*		*		*	
Selenium treatment											
Selenocystine	18.7 ± 0.09	15.0 ± 0.06	ab	67.5 ± 0.94	ab	9.0 ± 0.16	b	2.05 ± 0.08	a	10.2 ± 0.18	a
Sodium selenate	18.5 ± 0.13	15.3 ± 0.15	a	70.6 ± 1.25	a	10.3 ± 0.35	a	1.95 ± 0.12	a	9.8 ± 0.10	ab
Non-treated control	18.2 ± 0.05	14.4 ± 0.20	b	64.2 ± 1.18	b	8.7 ± 0.23	b	1.80 ± 0.09	b	9.3 ± 0.25	b
	n.s.										

n.s. not significant; * significant at *p* ≤ 0.05. Within each column, values followed by different lettersare statistically different according to Duncan’s test at *p* ≤ 0.05. Abbreviations: DM, dry matter; SS, soluble solids; TS, total sugars; MS, monosaccharides; TA, titratable acidity; TP, total proteins.

**Table 3 plants-08-00102-t003:** Shallot bulb macro-elemental composition as affected by microorganism inoculation and selenium treatments.

Experimental Factor	K		Ca		Mg	Na		P	S		Cl		NO_3_^−^	
Microorganism inoculation														
AMF formulate	8981 ± 850		1364 ± 135		1043 ± 28	986 ± 134		3078 ± 145	706 ± 42		1704 ± 98		716 ± 117	
Non-inoculated control	8195 ± 718		1003 ± 105		938 ± 52	935 ± 112		2716 ± 122	608 ± 46		1488 ± 85		551 ± 55	
	*		*		*	*		*	*		*		*	
Selenium treatment														
Selenocystine	9228 ± 304	a	1332 ± 207	a	1001 ± 27	1049 ± 47	a	2932 ± 124	702 ± 55	a	1663 ± 112	a	729 ± 114	a
Sodium selenate	9125 ± 581	a	1216 ± 176	a	997 ± 18	1057 ± 22	a	2845 ± 97	658 ± 51	ab	1615 ± 129	ab	667 ± 98	b
Non-treated control	7412 ± 294	b	1003 ± 158	b	974 ± 112	776 ± 9	b	2915 ± 382	612 ± 67	b	1511 ± 104	b	505 ± 37	c
					n.s.			n.s.						

n.s. not significant; * significant at *p* ≤ 0.05. Values followed by different letters are statistically different according to Duncan’s test at *p* ≤ 0.05.

**Table 4 plants-08-00102-t004:** Interaction between microorganism inoculation and selenium treatment on shallot bulb macro-elemental composition (mg·kg^−1^d.w.).

Element	Non-Inoculated						AMF Formulate					
	Non-Se		Selenocystine		Sodium Selenate		Non-Se		Selenocystine		Sodium Selenate	
K	7118 ± 594	c	8924 ± 931	ab	8544 ± 810	bc	7706 ± 789	bc	9531 ± 938c	a	9706 ± 873c	a
NO_3_	469 ± 35	d	615 ± 59	bc	569 ± 44	bc	541 ± 32	cd	842 ± 79c	a	765 ± 73c	a

Along each line, values followed by different letters are statistically different according to Duncan’s test at *p* ≤ 0.05.

**Table 5 plants-08-00102-t005:** Shallot bulb micro-elemental composition as affected by microorganism inoculation and selenium treatment.

Experimental Factor	Cu	Fe		Zn		Se		B		Mn	Si	
Microorganism inoculation												
AMF formulate	4.31 ± 0.28	65.4 ± 11.2		18.7 ± 3.4		4.66 ± 2.38		8.56 ± 1.11		881 ± 0.83	15.7 ± 1.2	
Non-inoculated control	3.75 ± 0.36	29.8 ± 2.9		12.4 ± 1.0		0.88 ± 0.57		6.84 ± 0.42		7.19 ± 1.03	11.8 ± 1.1	
	*	*		*		*		*		*		
Selenium treatment												
Selenocystine	4.10 ± 0.20	43.7 ± 15.9	b	16.1 ± 2.2	a	5.14 ± 3.64	a	7.47 ± 0.35	b	7.99 ± 0.02	14.4 ± 2.5	b
Sodium selenate	4.00 ± 0.12	40.9 ± 13.6	b	13.8 ± 1.49	b	3.03 ± 1.91	b	7.42 ± 0.23	b	8.16 ± 0.21	12.7 ± 2.7	c
Non-treated control	3.97 ± 0.76	58.2 ± 24.1	a	13.5 ± 2.5	b	0.14 ± 0.11	c	8.22 ± 2.01	a	7.85 ± 2.20	18.6 ± 5.24	a
	n.s.									n.s.		

n.s. not significant; * significant at *p* ≤ 0.05. Values followed by different letters are statistically different according to Duncan’s test at *p* ≤ 0.05.

**Table 6 plants-08-00102-t006:** Interaction between microorganism inoculation and selenium treatment on shallot bulb micro-elemental composition (mg·kg^−1^d.w.).

Element	Non-Inoculated Control						AMF Formulate					
	Non-Se		Selenocystine		Sodium Selenate		Non-Se		Selenocystine		Sodium Selenate	
B	6.21 ± 0.62	c	7.12 ± 0.71	b	7.19 ± 0.72	b	10.23 ± 1.02	a	7.81 ± 0.78	b	7.65 ± 0.76	b
Cu	3.21 ± 0.32	b	3.91 ± 0.39	ab	4.14 ± 0.41	a	4.73 ± 0.47	a	4.30 ± 0.43	a	3.90 ± 0.39	ab
Fe	34.1 ± 3.4	c	27.8 ± 2.8	c	27.3 ± 2.7	c	82.3 ± 8.2	a	59.5 ± 6.0	b	54.5 ± 5.5	b
Mn	5.65 ± 0.56	c	7.97 ± 0.80	b	7.95 ± 0.80	b	10.05 ± 1.00	a	8.01 ± 0.80	b	8.37 ± 0.84	ab
Zn	11.00 ± 1.10	d	13.91 ± 1.39	bc	12.33 ± 1.23	cd	15.92 ± 1.59	ab	18.28 ± 1.83	a	15.31 ± 1.53	ab
Si	13.36 ± 1.34	cd	11.87 ± 1.19	de	10.04 ± 1.00	e	23.84 ± 2.38	a	16.86 ± 1.69	b	15.36 ± 1.54	bc

Along each line, values followed by different letters are statistically different according to Duncan’s test at *p* ≤ 0.05.

**Table 7 plants-08-00102-t007:** Correlations between mineral elements in shallot bulbs.

	B	Cu	Fe	Mn	Zn	Si	Ca	K	Mg	Na	P	NO_3_
Cu	0.89	1										
Fe	0.88	0.69	1									
Mn	0.91 **	0.93 **	0.70	1								
Zn	0.60	0.70	0.71	0.63	1							
Si	0.89	0.64	0.97 **	0.66	0.60	1						
Ca	0.35	0.51	0.50	0.48	0.93 **	0.33	1					
K	−0.02	0.25	0.02	0.29	0.62	−0.16	0.85	1				
Mg	0.89	0.95 **	0.76	0.97 **	0.80	0.68	0.67	0.43	1			
Na	−0.21	0.16	−0.21	0.13	0.44	−0.40	0.71	0.95 **	0.27	1		
P	0.95 **	0.91 **	0.86	0.95 **	0.78	0.83	0.57	0.25	0.96 **	0.04	1	
NO_3_	0.08	0.29	0.26	0.25	0.80	0.077	0.96 **	0.92 **	0.46	0.83	0.33	
Al	−0.72	−0.76	−0.77	−0.80	−0.89	−0.62	−0.87	−0.63	0.91 **	0.45	−0.84	0.72

** significant at *p* ≤ 0.01.

**Table 8 plants-08-00102-t008:** Antioxidant compounds and activity of shallot bulbs as affected by microorganism inoculation and selenium treatment.

Experimental Factor	Polyphenols mg·g^−1^d.w.	Flavonoids mg Quercetin·g^−1^d.w.	Ascorbic Acid mg·g^−1^d.w.		AOA mg Gallic Acid·g^−1^d.w.
Microorganism inoculation					
AMF formulate	10.0 ± 0.12	0.23 ± 0.01	0.51 ± 0.01		17.5 ± 0.53
Non-inoculated control	9.7 ± 0.12	0.21 ± 0.01	0.45 ± 0.02		16.2 ± 0.47
	n.s.	n.s.	*		*
Selenium treatment					
Selenocystine	10.1 ± 0.19	0.22 ± 0.03	0.50 ± 0.03	a	17.1 ± 1.49
Sodium selenate	9.9 ± 0.18	0.23 ± 0.02	0.53 ± 0.01	a	16.8 ± 1.58
Non-treated control	9.6 ± 0.10	0.22 ± 0.03	0.43 ± 0.01	b	16.7 ± 1.26
	n.s.	n.s.			n.s.

n.s. not significant; * significant at *p* ≤ 0.05. Within each column, values followed by different lettersare statistically different accordingto Duncan’s test at *p* ≤ 0.05.

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
