# Peer review of "Effect of Selenium Biofortification and Beneficial Microorganism Inoculation on Yield, Quality and Antioxidant Properties of Shallot Bulbs"

_plants, 2019, doi:10.3390/plants8040102_

Round 1

Reviewer 1 Report

Review of the MS “Effect of Selenium Biofortification and Beneficial Microorganism Inoculation on Yield, Quality and Antioxidant Properties of Shallot Bulbs”

MS talks about the effect of the addition of Selenium and microorganism Inoculation on Yield, Quality and Antioxidant Properties of Shallot Bulbs

Abstract section: is clear and informative.

P 1 l 24-25 The flowing statement is not clear “Plant growth and yield were better affected by the application of selenocystine in mycorrhized plants and of selenate in the non-inoculated plants”.

Introduction section: is clear.

Materials and Methods section is clear.

P 3 L 106-107 The flowing statement is not clear: The AMF-based formulate Rhizotech Plus powder was placed at the bottom of each hole practiced into the soil prior to bulb planting.

P 3 L 124, space is missing: 20°C

P 4 L 163 space is missing: selenite(Se+4)

P 5 L 202 delete space: 70 %

Results and Discussion

P 5 L 2016-2017 The flowing statement is not clear: The mycorrhizal index, which expresses the percentage of root mycorrhizal colonization, did  not significantly change between the two determinations performed one month after the transplant  and at the crop cycle end.

P 6 L 254 Reference is missing: …of selenocystine treatment, whereas a similar effect was associated to the inorganic Se in potato.

Table 1: Why small and capital letters? Delete full stop in the middle of the sentence. Nin inoculated and non treated controls in mycorrhizal index are quite different? Please explain.

Figures 1-3: small and capital letters in the figures?

Letters in the Figure 3?

Conclusion section should better summarise the mail points of the MS!

Special comments

Delete space before %

Punctuations in the reference section are not equal.

MS bring interesting results! Well written MS!

I recommend minor revision.

Author Response

Thank you very much for your comments. We have made corrections according to your recommendations

Reviewer 2 Report

Interesting results, with many new data about internal quality of shallot. There is most of data positively affected by AMF fungi, just Im surprised that there were no year effect in field conditions observed.

How were selected bulbs for analyses - missing description.

I recommend to redesing figures, the minimum is to add confid. bars at least. Tables demands to add LSD or SE data next to means. I think you can use more compact tables to allow summing of data - e.g. what about to evaluate treatments effect on all beneficial compounds in bulbs?

Tab.4 - K content - strange, that all SE are 10 % of mean?

Glomus intraradices - use current name Rhizophagus.

cm - SI units..

Dates in methodology mentioned once (row 104, 108...) - does it mean it was the same date in both seasons?

I highly evaluate wide analytics. If you have also trace metals detected, some brief info can be put as well (if there were taken into consideration possible risk of contamination of samples).

Author Response

Thank you very much for your valuable comments. We revised the text according to your recommendations

Reviewer 3 Report

This paper describes a field study during which the quantitative and qualitative parameters of Allium were evaluated as the effect of arbuscular mycorrhiza inoculation in relation with selenium biofortification.

The experimental design is complex which includes several factors (fungi affect plant, Se affects plant and fungi, fungi affect Se uptake…). The experiments are comprehensive, and the results may be important from both agricultural and nutritional point of view. Before publication, Authors should extensively revise both the text and the figures. I found the visualization of the results confusing. My recommendations/comments/questions are the followings:

-there are no standard errors/deviations on the graphs. Information about the number of samples in case of each method is also missing (n=?). Please, provide these!

- on the graphs, the names of the samples are confusing. Control and non-treated control? Authors should rename their samples like non-inoculated or with AMF without AMF.

-the order of samples on the graphs is also confusing (at least for me). Usually, control/non-treated samples are the first then come the different treatments. I suggest to redesign all graphs in order to make them more clear.

-all graphs should have the same design. selenate or Na2SeO4?

-Authors speak about phytohormones in general. Which ones are relevant in this topic? Please specify them!

-despite the large dataset, the conclusion is short and weak. Please, provide more detailed conclusion possibly with some practical outcome.

English grammar and spelling need extensive revision.

Author Response

Thank you very much for your valuable comments. We have revised the manuscript according to your recommendations
